

# Consistency between GRUAN sondes, LBLRTM and IASI

Xavier Calbet[1], Niobe Peinado-Galan[2], Pilar Rípodas[1], Tim Trent[3,4], Ruud Dirksen[5], and Michael Sommer[5]

[1]AEMET; C/Leonardo Prieto Castro, 8; Ciudad Universitaria; 28071 Madrid; Spain
[2]University of Valencia; Physics Faculty; Carrer del Dr. Moliner, 50; 46100 Burjassot; Valencia; Spain
[3]Earth Observation Science; Department of Physics and Astronomy, University of Leicester, University Road, Leicester, LE1 7RH; United Kingdom
[4]National Centre for Earth Observation; Department of Physics and Astronomy, University of Leicester, University Road, Leicester, LE1 7RH, UK.
[5]Deutscher Wetterdienst; Meteorologisches Observatorium Lindenberg; Richard-Aßmann-Observatorium; Am Observatorium 12; 15848 Lindenberg/Tauche; Germany

*Correspondence to:* Xavier Calbet
xcalbeta@aemet.es

**Abstract.** Radiosonde soundings from the GRUAN data record are shown to be consistent with IASI measured radiances via the LBLRTM radiative transfer model in the part of the spectrum that is mostly affected by water vapour absorption in the upper troposphere (from 700 hPa up). This result is key to have consistency between radiosonde and satellite measurements for climate data records, since GRUAN, IASI and LBLRTM constitute reference measurements in each of their fields. This is specially the case for night time radiosonde measurements. Although the sample size is small (16 cases), day time GRUAN radiosonde measurements seem to have a small dry bias of 2.5% in absolute terms of relative humidity, located mainly in the upper troposphere, with respect to LBLRTM and IASI.

## 1  Introduction

Temperature and water vapour are two of the Essential Climate Variables (ECV) from Global Climate Observing System (GCOS). The ECVs are variables that are required to support the work of the United Nations Framework Convention on Climate Change (UNFCC) and the Intergovernmental Panel on Climate Change (IPCC) and which are technically and economically feasible for systematic observation. The required performance for satellite-based upper-air temperature and water vapour data products for climate from GCOS are very demanding (WMO GCOS , 2011). A summary of the requirements for atmospheric water vapour are shown in Table 1.

Temperature and water vapour are ECVs for which satellite observations can make a significant contribution; in particular from operational meteorological satellites by means of (passive) top of atmosphere (TOA) radiance measurements. Observations from space have several advantages; i) spatial coverage, which can be global and ii) continuous sampling of the atmosphere at regular intervals. Their main disadvantage is that they do not directly observe the Earth system, but indirectly by measuring the radiance from the Earth impinging on the satellite instrument. It is therefore mandatory to bridge the gap between satellite radiance measurements and ECVs. This is usually accomplished by modelling the pathways of radiation in



the atmosphere via Radiative Transfer Models (RTM). The inverse process allows for profiles of temperature and water vapour to be retrieved from the satellite measured radiances. This inversion can either be performed independently, or in the case of Numerical Weather Prediction (NWP) are assimilated into short and medium range forecasting models. The retrieval or assimilation method may contain inaccuracies either due to; i) imperfect modelling of the atmosphere, ii) auxiliary data used or iii)

inaccuracies inherent to the assumptions made by the technique itself, such as Gaussian uncertainties distribution assumptions or others.

For the measurements to be useful for climate or any other application they need to be adequately calibrated. The science of metrology defines best practices to achieve this goal. One key element in calibrating is traceability, by which various measurements can be compared. Metrological traceability is a property of a measurement result whereby the result can be related to a

reference through a documented unbroken chain of calibrations, each contributing to the measurement uncertainty. In simple terms metrological traceability is a direct link between the result of a measurement made in the field, and the result of the best possible measurement made in a calibration laboratory. It ensures that different measurement methods and instruments used at different locations and at different times produce reliable, repeatable, reproducible, compatible and comparable measurement results. When a measurement result is metrologically traceable, it can be confidently linked to the internationally accepted

measurement references. Traceability of metrological measurement results are assured by ensuring a documented, unbroken chain of instrument calibrations, from the operational instruments used for field measurements, all the way up the metrological hierarchy pyramid to the primary standard. At the top of the pyramid is an internationally defined and accepted reference, in most cases the International System of Units (SI), whose technical and organizational infrastructure has been developed by the Bureau International des Poids et Mesures – BIPM (www.bipm.org). For the case described here, the measurement process

consists of three fundamental elements; i) the radiance measurement from the satellite instrument, ii) the temperature and water vapour measurements from the radiosondes and iii) the RTM that establishes the link between them.

Throughout this measurement process, not all elements in the traceability chain are usually used comprehensively. In operational meteorological satellites, instruments are usually calibrated against well defined standards on the ground before launch. It is often the case that these instruments and particularly their components, have critical properties which vary with time,

degrading once the satellite is in space; effectively breaking the full traceability chain. RTM simulations of the observed TOA radiances usually do not propagate uncertainties arising from gaps in knowledge about the spectroscopy, therefore breaking again the traceability chain. Radiosonde measurements provided by the GCOS Reference Upper-Air Network (GRUAN)adhere to metrology best practices as they provide the accurate estimation of all uncertainties involved in the measurements (Dirksen et al., 2014).

With the goal of achieving an unbreakable chain of calibrations in the future, the satellite community is establishing a set of standards to which all other measurements can use as reference. The objective is to ultimately have these references calibrated through an unbroken traceability chain to primary standards. These current standards are described in the following:

– The Global Space-based Inter-Calibration System (GSICS) is an international collaborative effort initiated in 2005 by WMO (World Meteorological Organisation) and CGMS (Coordination Group for Meteorological Satellites) to monitor,

improve and harmonize the quality of observations from operational weather and environmental satellites of the Global



Observing System (GOS). GSICS aims at ensuring consistent accuracy among space-based observations worldwide for climate monitoring, weather forecasting and environmental applications. For infrared (IR)sensors, the standard instrument being adopted by GSICS is the Infrared Atmospheric Sounding Instrument (IASI) (GSICS , 2014; Hewison et al., 2013).

– For radiative transfer models the satellite community working with IR sensors commonly uses line-by-line radiative transfer models. They make use of laboratory measurements of gas absorption to perform its calculations, simulating the radiative transfer that occurs in the real atmopshere. One of such de-facto standards is LBLRTM (Line By Line Radiative Transfer Model), which is the one tested in this paper (Clough et al., 2005) .

– The GRUAN community takes great care of keeping the chain of traceability unbroken. The sonde data is provided
by GRUAN, removing, as far as possible, all the systematic errors in the measurements and quantifying very well the uncertainty in the measurements (WMO GCOS, 2013b).

When transforming IR measured radiances into atmospheric parameters (effectively performing what are known as a retrieval or data assimilation), it is necessary to keep the chain of traceability between all its elements unbroken. A first step into this direction is checking that all these elements are effectively consistent. That is the consistency between IASI measurements,
GRUAN sondes and LBLRTM calculations are a necessary condition to have an adequate chain of traceability. The consistency of all these components is the main subject of this paper.

Comparisons of measurements are usually done in temperature and humidity profile space, where a retrieval is compared to a radiosonde measurement (e.g. Tobin et al. (2006) or Reale et al. (2012)). Although being a legitimate comparison, this practice is not the best option when consistency is pursued. Retrieving a profile from a radiance spectrum is an ill-posed problem
which leads to solutions that are not unique. In other words, very different atmospheric profiles can lead to the same radiances measured at the top of the atmosphere. It is therefore much more convenient to perform the comparisons in radiance space, where the problem is uniquely determined (e.g. Calbet et al. (2011)). This is the practice followed in this paper. It is worth noting that there are two main disadvantages in using this technique. One is that an RTM to calculate the GRUAN derived radiances is needed for this exercise. This is not always the case when performing retrievals, in particular regression retrievals
based on real data (e.g. Blackwell  (2005)). The second one is that currently RTMs are precise and straight forward to use only in clear sky cases, and therefore the consistency study can only be practically done in clear sky scenes.

## 2  Consistency

In order for different components to be consistent, their measurements need to lie (on average) between their uncertainties. This is described by the Immler at el. (2010) equation

$$|m_1 - m_2| < k\sqrt{\sigma^2 + u_1^2 + u_2^2},$$  (1)





where $m_1$, $m_2$, $u_1$ and $u_2$ are the measurements and uncertainties from instrument 1 and 2 respectively. The term $\sigma$ is the uncertainty inherent in the particular comparison that is being performed. For the case of comparing IASI and GRUAN radiosonde data, the biggest component in this $\sigma$ term is usually the collocation uncertainty. The $k$ parameter is a value that estimates the ratio between both sides of the inequation. For the measurements to be consistent, this $k$ value has to be around

two (Immler at el., 2010). If the measurements lie within their associated uncertainties (i.e. $\sqrt{u_1^2 + u_2^2}$), then the collocation uncertainty can be assumed to be small. This is the ideal situation when validating IASI retrievals with radiosondes (Calbet , 2016).

The different components that are verified in this paper to be consistent are described below:

## 2.1   IASI

Space-borne IR hyperspectral instruments typically measure Earth views in a spectral range from 600 to $3000\,\mathrm{cm^{-1}}$ wavenumbers with a spectral sampling of about $0.25\,\mathrm{cm^{-1}}$ providing thousands of channels across their full spectral range. The typical noise per channel of these instruments is roughly in the range from 0.1 to 0.8 K as noise equivalent delta temperature at 280 K. From these measurements it is possible to retrieve atmospheric profiles of temperature and water vapour with a relatively high vertical resolution and high degree of accuracy. These, so called, retrievals can have a temperature accuracy of about 1 K in

layers 1 km thick and humidity accuracy from 10 to $20\%$ in layers 2 km thick within the troposphere (Smith et al., 2001). One of such IR hyperspectral instrument is IASI, described by Chalon et al. (2001) and Blumstein et al. (2004). It is a Fourier transform spectrometer currently on board the polar orbiting satellites Metop-A and Metop-B. IASI is measuring within the whole spectral range from 645 to $2760\,\mathrm{cm^{-1}}$ with a spectral sampling of $0.25\,\mathrm{cm^{-1}}$, an apodized effective resolution of $0.5\,\mathrm{cm^{-1}}$ and with a spatial resolution of about 12 km at nadir. Its overall measurement uncertainty has been determined by CNES, who

has derived the IASI covariance matrix instrument measurement uncertainty (Pequignot et al., 2008).

IASI has been compared with various calibration references, both pre-flight and in-orbit. However, reference values with associated uncertainties that are traceable to SI standards have not been assigned. Moreover, while in-orbit the instrument has no SI source and hence the traceability to an SI standard once the satellite is launched is lost. Despite this, due to its quality and long term radiometric stability the GSICS community has declared IASI as a standard to which all other IR satellite sensors

can reference to (Hewison et al., 2013).

## 2.2   LBLRTM

Accurate spectra at the top of the atmosphere were generated using the Line By Line Radiative Transfer Model (LBLRTM, Clough et al. (2005)). LBLRTM has a long development history and for the current study one of the latest versions (12.2) was adopted. LBLRTM is a versatile highly accurate radiation code which describes the interaction between matter and radiation

at a single wavenumber. Its spectral resolution for this particular application lies bewteen 0.00025 and $0.0005\,\mathrm{cm^{-1}}$. The accuracy of LBLRTM has been demonstrated in several publications (e.g. Tjemkes et al. (2003)). LBLRTM is considered as a standard by the IR RTM community.





## 2.3 GRUAN

GCOS has established and is continuing to develope a reference network for upper-air climate observations (GRUAN). GCOS is a joint undertaking of the World Meteorological Organization (WMO), the Intergovernmental Oceanographic Commission (IOC) of the United Nations Educational Scientific and Cultural Organization (UNESCO), the United Nations Environment

Programme (UNEP) and the International Council for Science (ICSU). Its goal is to provide comprehensive information on the total climate system, involving a multidisciplinary range of physical, chemical and biological properties, and atmospheric, oceanic, hydrological, cryospheric and terrestrial processes.

GRUAN is a ground-based network for reference observations of upper-air climate parameters. GRUAN is expected to provide long-term, highly accurate measurements of atmospheric profiles; complemented by ground-based state of the art

instrumentation to constrain and calibrate data from more spatially-comprehensive global observing systems (inc. satellites and current radiosonde networks). The primary goal is to fully characterize the properties of the atmospheric column and their changes. GRUAN is envisaged as a network of 30-40 high-quality, long-term, upper-air observing stations, building on existing observational networks.

The data that is currently certified within the GRUAN standards is the Vaisala RS92 radiosonde data, which is the data that

will be used in this paper. The specific GRUAN data used in this paper is the "RS92 GRUAN Data Product Version 2", which has the "RS92-GDP.2" key (Sommer et al., 2012). The GRUAN data processing for the RS92 radiosonde was developed to meet the criteria as a reference measurement (Dirksen et al., 2014). These criteria stipulate the collection of metadata, the use of well-documented correction algorithms, and estimates of the measurement uncertainty. An important and novel aspect of the GRUAN processing is that the uncertainty estimates (random and systematic components) are vertically resolved.

## 3  Methodology

### 3.1  Data Selection

In order to verify the consistency of all the elements involved in the comparison, ideally a collocation uncertainty close to zero is desired ($\sigma \approx 0$, Eq. 1). Pougatchev et al. (2009) studied the variability of temperature and water vapour with radiosondes launched from Lindenberg reaching the conclusion that to minimize the collocation uncertainty a spatial and temporal window

of 25 km and 30 minutes respectively is needed. Although this collocation criteria initially seemed sufficient, during the development of this study it was noted that for water vapour these criteria (in reality) are too relaxed. Therefore, even stricter criteria are needed (see section 4 and 5 for a discussion on this).

The IASI instrument flies on board of Metop which is in a mid-morning orbit, overpassing the equator at around 09:00 hours local solar time. Since the GRUAN radiosondes are mostly launched at synoptic times (00Z and 12Z), the locations on

the globe where IASI and the GRUAN radiosondes would coincide are located over the middle of the Atlantic or the western Pacific (Fig. 1). As a consequence the only GRUAN station that meets this criteria is the one located on the island of Manus





in the tropical western Pacific region. It should be noted that this station has been discontinued and is no longer providing any data to GRUAN. The time interval within which GRUAN data is available for Manus ranges from 2011 to 2013.

Radiative transfer models are in practice accurately characterized for clear sky cases, making it therefore necessary to select the clear sky scenes. There are a total of 597 coincident IASI overpasses and GRUAN radiosonde launches over Manus during

this period. From these a further selection of clear sky cases is needed. The cloud flag available in the standard IASI L1c product is used for a first screening, leaving 76 clear cases. To perform the radiation matching between GRUAN derived and IASI radiances a perfectly clear sky scene is needed. Since the IASI L1c cloud flag does not have an efficiency of 100% in detecting clear cases, a further visual screening of the scenes as seen by AVHRR (Advanced Very High Resolution Radiometer) have been performed. This instrument is flown on board of the same satellite (Metop) and has the advantage of having a much

higher spatial resolution of around 1 km at nadir, which makes it specially useful for cloud detection. After this second clear sky screening is done only 27 cases are left. These cases are the ones used in the remaining of this paper. All cases where a GRUAN and IASI collocation over Manus which are clear according to the IASI L1c cloud fraction are listed in Table 2.

## 3.2   Further processing of the GRUAN profiles

According to Calbet et al. (2011) one of the key subjects identified as critical to match IASI radiances to the ones based on

RS92 radiosonde data is the radiation dry bias correction applied to the radiosonde humidity measurements. These corrections are needed in the RS92 data to realistically represent the water vapour present in the atmosphere. The standard processing of the radiosonde data made by GRUAN (Dirksen et al., 2014) corrects for this effect and no further processing is needed.

The useability of the RS92 humidity profiles is largely determined by the amount of water vapor present. Above the tropopause the water vapor level drops by approximately 2 orders of magnitude. The intrinsic uncertainty of the radiosonde

humidity profile is 1% RH or more, meaning that at low relative humidity levels, which typically occur in the stratosphere, the relative uncertainty of the measurement is 100%, which renders the data of little use in the present exercise. In the examples in this paper, humidity measurements from the GRUAN radiosondes are taken as useful when they are below 100 hPa, which, for these cases, is just below the tropopause. Regarding temperature, the burst of the balloon is what limits their altitude. The GRUAN objective is to aim for a maximum altitude of 5 hPa. For thicker balloons, in the range of 600 to 1200 grams, the burst

of the balloons reaches heights between 10 to 4 hPa. For radiosondes launched from Manus they are typically limited to an altitude between 30 and 10 hPa due to the use of thinner balloons. This would then be the limit for temperature measurements of this GRUAN data. Because of these upper limitations on temperature and humidity measurements and in order to be able to apply the radiative transfer to the radiosonde profiles, it is necessary to extend them above this altitude up to the TOA. This is done by complementing them in this upper region with ECMWF fields, by taking the nearest operational analysis to the

radiosonde launch location in space and time.

The RS92 sensor measures the relative humidity of the ambient air, whereas the RTM needs as input the water vapour concentration, typically specific humidity. It is therefore necessary to convert the humidity measurements from relative humidity to specific humidity. To do this, a water vapour saturation curve is needed. The final calculated radiances, especially for channels which are most sensitive to upper air regions such as the high troposphere or which have low water vapour concentrations,





such as the ones used in this paper, is very much dependent on the type of formulation which is selected (Murphy and Koop , 2005). For consistency reasons and also considering that the GRUAN community takes as practical the Hyland and Wexler (1983) curve, this is the one used in this paper.

Finally, the radiosonde profiles are smoothed with a mean filter of 100 points in the vertical. The reason for this is that the

original radiosonde data exhibits high oscillations and spikes which are either spurious or too noisy and it is therefore not recommended to feed this raw data as input to the RTM. It must be considered that in any case, IASI measured radiances or retrievals are not sensitive to particular small scales in the vertical.

Figure 2 illustrates the processing performed on the GRUAN profiles to be able to serve them as input to the RTM.

### 3.3 RTM radiance calculations and their uncertainties

Once the profiles are prepared, they are used as input to LBLRTM. To avoid surface effects in the calculated radiances, only the higher absorptive water vapour channels are used in this study. The channels used range from $1400$ to $1900\,\mathrm{cm}^{-1}$, covering practically all atmospheric levels from around $700\,\mathrm{hPa}$ and above. Figure 3 shows calculated radiance differences for a particular atmospheric profile. The output of LBLRTM are radiances at a very high spectral resolution. This spectra has to then be modified to IASI specifications. To do this, the spectra are smoothed down to IASI spectral resolution using the IASI

spectral response function (SRF). Finally a calculated spectra is obtained with the complete characteristics of an ideal IASI instrument.

The radiosonde profile uncertainties provided by GRUAN (Dirksen et al., 2014) are propagated into radiance space to determine whether all measurements are compatible (Eq. 1). The uncertainties provided with the GRUAN measurements are defined on a per radiosonde level basis and there are no covariance terms between levels. These covariances are critical in the

propagation of the uncertainties from profile into radiance space. This is physically due to the fact that IASI observes the Earth viewing all atmospheric levels at the same time.

There are several ways to propagate the uncertainties from atmospheric profile into radiance space. The most straight forward way of propagating uncertainties is by using the parameter derivatives. In this case, the Jacobians of the radiances with respect to the atmospheric profiles from the radiative transfer equations could be multiplied to the atmospheric profile uncertainties to

obtain the radiance uncertainties. These Jacobians are usually available as an output of the RTM. Due to the large number of IASI spectral points and the number of levels in the GRUAN profiles, this method is computationally expensive and impractical for this study. Also, the Jacobian of the radiances is needed, which for the case of LBLRTM it can be quite impractical to use and obtain. Added to this the fact that the uncertainty covariances between levels is not available for GRUAN profiles, it is not evident how to use the Jacobians for this purpose. In this paper, a more practical approach has been taken. The uncertainty

propagation has been performed assuming two extreme cases: uncertainty is completely uncorrelated between levels and there is a perfect correlation between uncertainties from all levels. Therefore, the truth most likely lies in between these two extremes.

To propagate the uncertainties (assuming no uncertainty correlation between levels), a Monte Carlo method was applied. For each level and variable a random perturbation is added; having a Gaussian distribution with zero bias, and a standard deviation equal to the corresponding GRUAN global uncertainty on that level. Each level is perturbed totally independently





from the next. After this perturbation is applied, the radiances at the top of the atmosphere are calculated using LBLRTM. This process is repeated several times to obtain the standard deviation of the radiances within the Monte Carlo approach. This final standard deviation is taken as the uncertainty of the GRUAN profiles in radiance space. One result for a particular profile is shown in Figure 5 as an orange curve. It is worth noting that the resulting radiance uncertainty is small compared to the overall

IASI instrument uncertainty. The reason for this lies in the lack of any uncertainty correlation between levels which ends up compensating the perturbation in radiance space from one level with the one from another level.

The propagation of uncertainties when assuming a perfect correlation of uncertainties between levels, is done by perturbing the temperature and humidity variables by plus or minus the uncertainty as given by GRUAN from that parameter and level consistently over the complete profile. In other words if the temperature is perturbed by plus one GRUAN uncertainty at the

surface, the rest of the temperature profile is also perturbed by plus one GRUAN uncertainty for each level. Therefore, there are a total of four different profiles; two coming from the plus and minus addition of one GRUAN uncertainty times another two coming from the two variables, temperature and water vapour. Radiances are then calculated for these four profiles using LBLRTM. To derive a radiance uncertainty from these calculations, all four calculated radiances are subtracted pairwise giving a total of six differences. Of these six, the greatest difference is taken as the final uncertainty for uncertainty correlated levels.

The combination that provides the greatest uncertainty in this case consisted of plus one GRUAN uncertainty in temperature and minus one GRUAN uncertainty in humidity. Results are shown in Figure 5 as a green curve. Note how this uncertainty is much greater than the previously calculated uncertainty with no uncertainty correlation between levels, as it would be expected.

## 4   Comparisons

The differences between calculated radiances obtained from the results of LBLRTM applied to the GRUAN radiosondes, and

the IASI measured radiances are computed for the comparison. For illustrative purposes, the calculated radiances obtained from the nearest in space and time ECMWF operational analysis profile are also compared to IASI. It is worth recalling that all cases analysed in this paper are clear scenes. Figure 3 illustrates one such sample. The red curve indicates the GRUAN radiosonde calculated radiances compared to IASI. The thickness of this red line indicates the uncertainty in the radiances obtained using the Monte Carlo method and assuming there is no uncertainty correlation between levels. This thickness is so

small that is difficult to distinguish in the Figure. The blue curve shows the ECMWF profile calculated radiances compared to IASI measured ones. The black line indicates the overall IASI instrument uncertainty. As we can see for this case, the match is quite remarkable both for GRUAN and ECMWF. Both radiance differences fall overall within the IASI instrument uncertainty (black line).

Figure 4 illustrates another sample, again, under a clear scene. In this case the match is quite poor. Neither the GRUAN

radiosonde nor the ECMWF profile matches the IASI radiances well. We firmly believe that the main cause for this is the extremely high variability of water vapour in the atmosphere, which makes the perfect collocation of GRUAN radiosondes and ECMWF profiles with IASI very difficult. In other words, the $\sigma$ term in Eq. 1 is significant. Note that this is in contrast with Calbet et al. (2011) where all cases did match individually. The main difference with respect to this study resides is





that in Calbet et al. (2011) dual radiosonde consecutive launches one hour apart were available, making a time interpolation possible. Whereas, in this paper, the time interpolation is impossible due to only a single radiosonde sounding available per IASI collocation.

To overcome this issue the average of the radiance difference of different cases was calculated. The expectation is that the random perturbations due to collocation uncertainties would average out. For this to happen, these perturbations need to have a normal random distribution. Results are shown in Figure 6 for the night time cases, where it can be seen that the average difference effectively lies within uncertainty values. In this figure, the average of the difference between measurements ($m_1$ and $m_2$ in Eq. 1) lie within the addition of uncertainties of the measurements ($u_1$ and $u_2$ in Eq. 1), which are represented in this figure as a black line for the IASI overall instrument uncertainty and as the thickness of the red line for the GRUAN uncertainty (assuming no uncertainty correlation between levels). The dotted line indicates two times the composition of both instrument uncertainties, which would be the $k\sqrt{u_1^2 + u_2^2}$ term in Eq. 1. This is the proof that GRUAN, LBLRTM and IASI are indeed consistent with a $k \approx 1$ from Eq. 1. In the same figure it can also be verified that ECMWF behaves similarly. The few channels that clearly lie outside the overall IASI instrument uncertainty in Figure 6 are due to the fact that these channels, wavenumbers below $1500\,\mathrm{cm}^{-1}$ and around $1585\,\mathrm{cm}^{-1}$, are affected by surface effects that are not adequately modelled here. For other channels which also lie outside the uncertainty ranges, wavenumbers between $1800\,\mathrm{cm}^{-1}$ and $1840\,\mathrm{cm}^{-1}$, the reason is unknown.

The standard deviation of the differences for all samples are shown in Figure 5 as a red curve for GRUAN and as a blue curve for ECMWF. These curves indicate the total uncertainty in the comparison, including collocation, instrument and RTM uncertainties.

Figure 7 shows the day time cases. In this example the coincidence is not satisfactory, lying some parts of the spectra outside of the uncertainty tolerances. This is not the case for ECMWF, which does lie well within the uncertainties (like in the night time cases). This is a clear indication that GRUAN data seems to suffer from a slight bias in the day time measurements. To quantify this bias, further calculations were made where the relative humidity from the GRUAN radiosondes was artificially incremented by adding 2.5% in absolute terms of relative humidity. This result is shown in Figure 8. The match here is reasonable such that these radiances show that GRUAN day time radiosondes seem to have a dry bias of 2.5%. Although 2.5% of relative humidity was added to the complete radiosonde profile, the IASI channels that are being analysed here are mostly sensitive to the upper tropospheric water vapour (from 700hPa up). Therefore, the bias is mostly coming from these upper layers.

It is interesting to note how the sample size shrinks as we select the data more and more. The initial number of collocations of IASI with GRUAN over Manus during the period this station was operational (2011–2013) was of 597 cases. Once only clear cases are selected, following the cloud flag present in the IASI L1 product, 76 cases are left. After visual inspection of the scenes, to remove potential residual cloudy cases, only 27 cases remain. Of these, 11 cases are measured during night time, which are the ones that provide a good match between IASI and GRUAN, and the other 16 day time cases do not provide a reasonable match up. This stresses the need for having high quality radiosonde observations, such as those provided by GRUAN, collocated with satellite overpasses.



## 5 Conclusions

It has been verified that GRUAN, LBLRTM and IASI are indeed consistent with each other. This is the main result of this paper. This is a key finding when using these measurements in fields where a high accuracy is needed like climate science. Even though the consistency between GRUAN and IASI cannot be proven on cloudy scenes, it can be expected that GRUAN quality remains unchanged under any conditions, serving its main purpose as a reference network for climate and other applications. Consistency is also necessary for applications such as obtaining accurate retrievals from IASI measurements (Calbet , 2016). It is not straight forward to reach this result and many critical issues have been identified, these are:

– Adequate collocations are needed. Scale lengths and times of water vapour are extremely small as Carbajal Henken et al. (2015) have clearly demonstrated using MERIS data. This makes it very complicated to obtain perfect match ups. If a small collocation uncertainty is desired, it is mandatory to use small collocation windows (typically smaller than 25 km and 30 min). Also desirable would be a double radiosonde launch, where both radiosondes are launched separated by approximately one hour. In this way, a time interpolation known as Tobin interpolation is possible (Tobin et al., 2006). This technique provided match ups even for individual cases in the past (Calbet et al., 2011). Also, standard deviations of the complete sample were very close to the IASI instrument uncertainty. This result is very clear in Fig. 15 of Calbet et al. (2011), as opposed to the results obtained in this paper with single radiosonde launches (red curve of Figure 5).

– The water vapour saturation function used to convert from relative humidity measured by the radiosonde to some form of water concentration such as specific humidity is highly critical. In this case, following Dirksen et al. (2014), the Hyland and Wexler  (1983) water vapour saturation function was used.

– It is also very important to correct the RS92 radiosonde measurements from all potential systematic errors it might have. For this, the GRUAN processing plays a key role removing such biases and providing the necessary uncertainties to make a meaningful comparison.

– Proper cloud detection is also critical. A few cases with spurious clouds will kill the consistency results. In this paper, an additional visual cloud detection was done on the data with the help of AVHRR images.

– GRUAN processing seems to still have a remaining bias of around 2.5% in absolute terms of relative humidity for radiosondes flown during day time, which is corroborated by the fact that this effect does not seem to show up in night time sondes nor in ECMWF profiles.

– Results from this paper are drawn with very limited sample sizes (11 night time and 16 day time), so they should be taken with care. A study with more cases should be performed in the future. It should also be stressed the need for more radiosonde launches coincident with satellite overpasses.

– The results shown in this paper would have been impossible with other data of lower quality than GRUAN. The fact that the GRUAN community strives for providing bias free data and an uncertainty associated with each measurement is what has made this study possible.



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



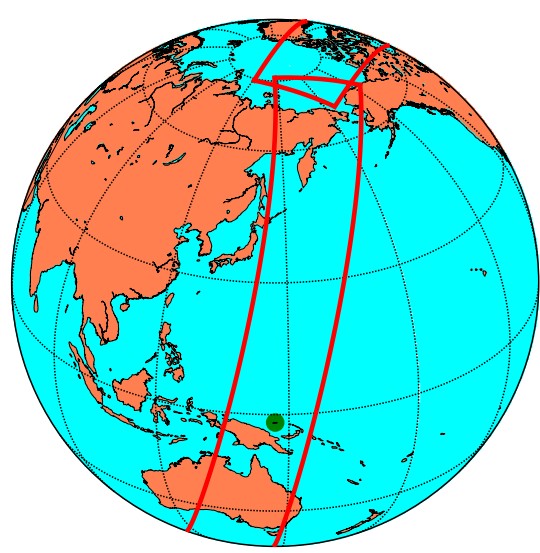

**Figure 1.** IASI complete orbit (red) on 2011/11/04 at 23:20:57 Z over the observatory location, Manus island (green dot).



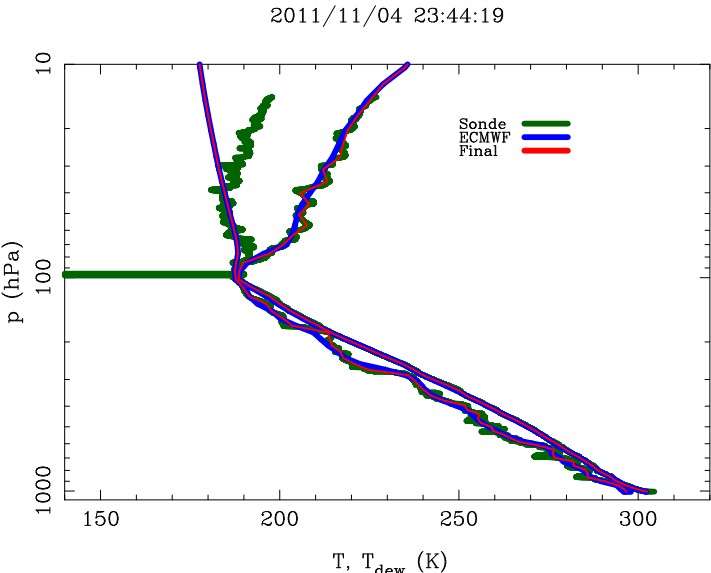

**Figure 2.** Individual sample of a raw GRUAN sonde (green), ECMWF profile (blue) and the final profile after pre–processing before being fed as input to LBLRTM (red). The red, green and blue lines to the right show the temperature profiles, while the ones to the left show the hunmidity profiles represented as dew point temperature.

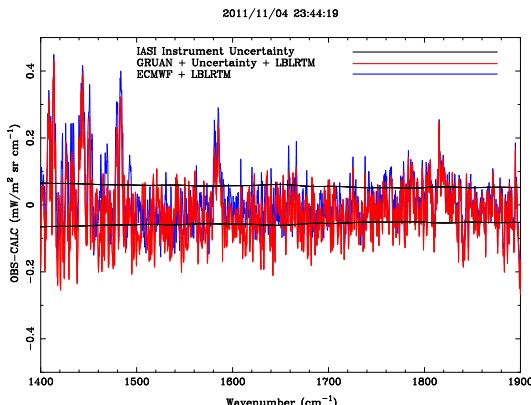

**Figure 3.** IASI observed minus calculated radiances (OBS-CALC) for a sample sonde (2011/11/04 23:44:19Z). Calculated radiances derived from LBLRTM and GRUAN sondes (red) and ECMWF (blue). IASI overall instrument uncertainty (black). The thickness of the red line denotes the GRUAN uncertainty propagated into radiance space assuming no uncertainty correlation between levels.




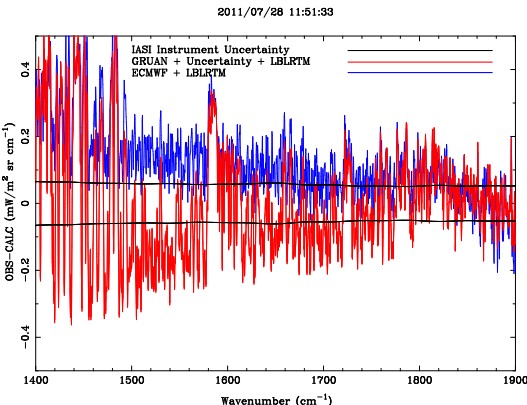

**Figure 4.** IASI observed minus calculated radiances (OBS-CALC) for a sample sonde (2011/07/28 11:51:33Z). Calculated radiances derived from LBLRTM and GRUAN sondes (red) and ECMWF (blue). IASI overall instrument uncertainty (black). The thickness of the red line denotes the GRUAN uncertainty propagated into radiance space assuming no uncertainty correlation between levels.

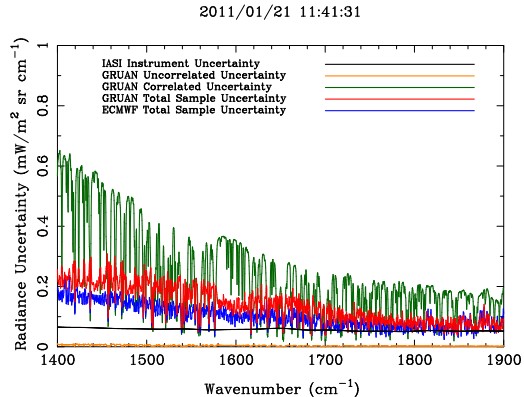

**Figure 5.** Several radiance uncertainties. IASI overall instrument uncertainty (black); GRUAN instrument uncertainty propagated into radiance space assuming no uncertainty correlation between levels for the 2011/01/21 11:41:31 case (orange); GRUAN instrument uncertainty propagated into radiance space assuming perfect uncertainty correlation between levels for the 2011/01/21 11:41:31 case (green); calculated radiance standard deviation from GRUAN sondes for all the completely clear scenes and night time cases (red); calculated radiance standard deviation from ECMWF profiles for all the clear scenes and night time cases (blue).



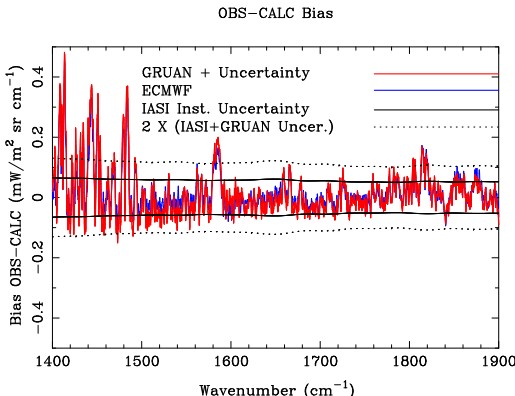

**Figure 6.** Average radiance difference (bias) between IASI observed and calculated radiances for the completely clear scenes and night time cases. Calculated radiances are derived from GRUAN sondes (red) and ECMWF profiles (blue). The thickness of the red line denotes the GRUAN uncertainty propagated into bias radiance space assuming no uncertainty correlation between levels. The dotted black line indicates two times the square root of the squares of the IASI overall instrument plus GRUAN uncertainties (the $k\sqrt{u_1^2 + u_2^2}$ term in Eq. 1).

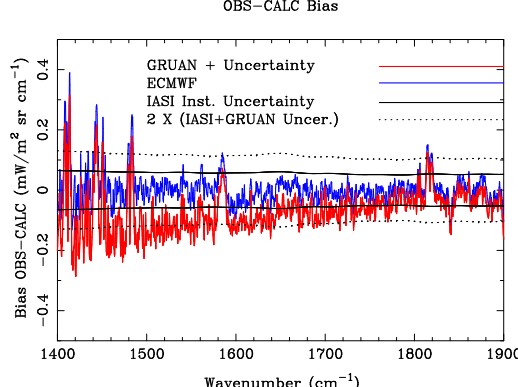

**Figure 7.** Average radiance difference (bias) between IASI observed and calculated radiances for the completely clear scenes and day time cases. Calculated radiances are derived from GRUAN sondes (red) and ECMWF profiles (blue). The thickness of the red line denotes the GRUAN uncertainty propagated into bias radiance space assuming no uncertainty correlation between levels. The dotted black line indicates two times the square root of the squares of the IASI overall instrument plus GRUAN uncertainties (the $k\sqrt{u_1^2 + u_2^2}$ term in Eq. 1).



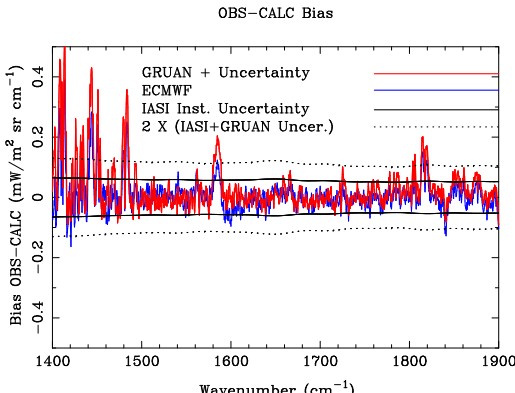

**Figure 8.** Same as Fig. 7 but artificially adding 2.5% in absolute terms of relative humidity to the complete GRUAN sonde profile before calculating its radiances.

**Table 1.** GCOS target requirements for the satellite–based Essential Climate Variable (ECV) of water vapour (WMO GCOS , 2011).

| Variable/ Parameter | Horizontal Resolution | Vertical Resolution | Temporal Resolution | Accuracy | Stability |
|---|---|---|---|---|---|
| Total column-water vapour | 25 km | N/A | 4 h | 2% | 0.3% |
| Tropospheric and lower-stratospheric profiles of water vapour | 25 km (troposphere) 100 - 200 km (stratosphere) | 2 km | 4 h (troposphere) daily (stratosphere) | 5% | 0.3% |
| Upper-tropospheric humidity | 25 km | N/A | 1 h | 5% | 0.3% |

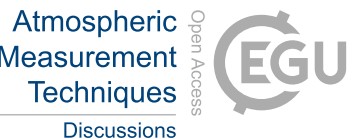

**Table 2.** GRUAN and IASI collocation cases over Manus, where only the clear cases according to IASI L1c cloud fraction are listed.

| # | Place | Type | GDP | Date | UTC | Clear IASI | Clear Visual | Day/Night | # | Place | Type | GDP | Date | UTC | Clear IASI | Clear Visual | Day/Night |
|---|---|---|---|---|---|---|---|---|---|---|---|---|---|---|---|---|---|
| 01 | Manus | RS92 | 2 | 10.01.2011 | 12:00:00 | Yes | No | Day | 39 | Manus | RS92 | 2 | 24.12.2012 | 00:00:00 | Yes | No | Night |
| 02 | Manus | RS92 | 2 | 11.01.2011 | 00:00:00 | Yes | Yes | Night | 40 | Manus | RS92 | 2 | 18.01.2013 | 00:00:00 | Yes | No | Night |
| 03 | Manus | RS92 | 2 | 21.01.2011 | 12:00:00 | Yes | No | Day | 41 | Manus | RS92 | 2 | 19.01.2013 | 00:00:00 | Yes | No | Night |
| 04 | Manus | RS92 | 2 | 24.01.2011 | 00:00:00 | Yes | No | Night | 42 | Manus | RS92 | 2 | 22.01.2013 | 00:00:00 | Yes | No | Night |
| 05 | Manus | RS92 | 2 | 28.06.2011 | 00:00:00 | Yes | Yes | Night | 43 | Manus | RS92 | 2 | 05.03.2013 | 12:00:00 | Yes | Yes | Day |
| 06 | Manus | RS92 | 2 | 10.07.2011 | 12:00:00 | Yes | Yes | Day | 44 | Manus | RS92 | 2 | 19.03.2013 | 12:00:00 | Yes | No | Day |
| 07 | Manus | RS92 | 2 | 28.07.2011 | 12:00:00 | Yes | Yes | Day | 45 | Manus | RS92 | 2 | 30.03.2013 | 00:00:00 | Yes | Yes | Night |
| 08 | Manus | RS92 | 2 | 31.07.2011 | 00:00:00 | Yes | Yes | Night | 46 | Manus | RS92 | 2 | 21.04.2013 | 12:00:00 | Yes | No | Day |
| 09 | Manus | RS92 | 2 | 04.09.2011 | 00:00:00 | Yes | No | Night | 47 | Manus | RS92 | 2 | 22.04.2013 | 00:00:00 | Yes | No | Night |
| 10 | Manus | RS92 | 2 | 15.10.2011 | 12:00:00 | Yes | Yes | Day | 48 | Manus | RS92 | 2 | 01.05.2013 | 12:00:00 | Yes | Yes | Day |
| 11 | Manus | RS92 | 2 | 18.10.2011 | 00:00:00 | Yes | No | Night | 49 | Manus | RS92 | 2 | 02.05.2013 | 00:00:00 | Yes | No | Night |
| 12 | Manus | RS92 | 2 | 19.10.2011 | 12:00:00 | Yes | Yes | Day | 50 | Manus | RS92 | 2 | 16.05.2013 | 12:00:00 | Yes | No | Day |
| 13 | Manus | RS92 | 2 | 26.10.2011 | 00:00:00 | Yes | Yes | Night | 51 | Manus | RS92 | 2 | 17.05.2013 | 00:00:00 | Yes | No | Night |
| 14 | Manus | RS92 | 2 | 31.10.2011 | 00:00:00 | Yes | No | Night | 52 | Manus | RS92 | 2 | 19.05.2013 | 12:00:00 | Yes | No | Day |
| 15 | Manus | RS92 | 2 | 05.11.2011 | 00:00:00 | Yes | Yes | Night | 53 | Manus | RS92 | 2 | 20.05.2013 | 12:00:00 | Yes | No | Day |
| 16 | Manus | RS92 | 2 | 15.11.2011 | 00:00:00 | Yes | No | Night | 54 | Manus | RS92 | 2 | 24.05.2013 | 12:00:00 | Yes | No | Day |
| 17 | Manus | RS92 | 2 | 20.11.2011 | 00:00:00 | Yes | No | Night | 55 | Manus | RS92 | 2 | 25.05.2013 | 00:00:00 | Yes | Yes | Night |
| 18 | Manus | RS92 | 2 | 18.12.2011 | 00:00:00 | Yes | No | Night | 56 | Manus | RS92 | 2 | 25.05.2013 | 12:00:00 | Yes | Yes | Day |
| 19 | Manus | RS92 | 2 | 23.12.2011 | 00:00:00 | Yes | Yes | Night | 57 | Manus | RS92 | 2 | 26.05.2013 | 00:00:00 | Yes | No | Night |
| 20 | Manus | RS92 | 2 | 29.12.2011 | 00:00:00 | Yes | No | Night | 58 | Manus | RS92 | 2 | 31.05.2013 | 00:00:00 | Yes | No | Night |
| 21 | Manus | RS92 | 2 | 23.01.2012 | 12:00:00 | Yes | No | Day | 59 | Manus | RS92 | 2 | 05.06.2013 | 00:00:00 | Yes | No | Night |
| 22 | Manus | RS92 | 2 | 27.01.2012 | 00:00:00 | Yes | Yes | Night | 60 | Manus | RS92 | 2 | 26.06.2013 | 12:00:00 | Yes | No | Day |
| 23 | Manus | RS92 | 2 | 19.02.2012 | 00:00:00 | Yes | No | Night | 61 | Manus | RS92 | 2 | 06.07.2013 | 00:00:00 | Yes | No | Night |
| 24 | Manus | RS92 | 2 | 13.04.2012 | 00:00:00 | Yes | Yes | Night | 62 | Manus | RS92 | 2 | 26.07.2013 | 00:00:00 | Yes | No | Night |
| 25 | Manus | RS92 | 2 | 08.05.2012 | 00:00:00 | Yes | No | Night | 63 | Manus | RS92 | 2 | 03.08.2013 | 00:00:00 | Yes | No | Night |
| 26 | Manus | RS92 | 2 | 17.05.2012 | 00:00:00 | Yes | No | Night | 64 | Manus | RS92 | 2 | 04.08.2013 | 00:00:00 | Yes | No | Night |
| 27 | Manus | RS92 | 2 | 18.05.2012 | 00:00:00 | Yes | No | Night | 65 | Manus | RS92 | 2 | 06.08.2013 | 12:00:00 | Yes | Yes | Day |
| 28 | Manus | RS92 | 2 | 30.06.2012 | 12:00:00 | Yes | No | Day | 66 | Manus | RS92 | 2 | 17.08.2013 | 00:00:00 | Yes | Yes | Night |
| 29 | Manus | RS92 | 2 | 15.07.2012 | 12:00:00 | Yes | No | Day | 67 | Manus | RS92 | 2 | 09.09.2013 | 12:00:00 | Yes | Yes | Day |
| 30 | Manus | RS92 | 2 | 17.08.2012 | 00:00:00 | Yes | Yes | Night | 68 | Manus | RS92 | 2 | 21.09.2013 | 00:00:00 | Yes | No | Night |
| 31 | Manus | RS92 | 2 | 19.09.2012 | 00:00:00 | Yes | No | Night | 69 | Manus | RS92 | 2 | 23.09.2013 | 00:00:00 | Yes | No | Night |
| 32 | Manus | RS92 | 2 | 25.09.2012 | 00:00:00 | Yes | Yes | Night | 70 | Manus | RS92 | 2 | 27.09.2013 | 00:00:00 | Yes | No | Night |
| 33 | Manus | RS92 | 2 | 03.10.2012 | 00:00:00 | Yes | No | Night | 71 | Manus | RS92 | 2 | 23.10.2013 | 00:00:00 | Yes | Yes | Night |
| 34 | Manus | RS92 | 2 | 15.10.2012 | 12:00:00 | Yes | Yes | Day | 72 | Manus | RS92 | 2 | 04.11.2013 | 00:00:00 | Yes | No | Night |
| 35 | Manus | RS92 | 2 | 18.10.2012 | 00:00:00 | Yes | No | Night | 73 | Manus | RS92 | 2 | 16.11.2013 | 00:00:00 | Yes | No | Night |
| 36 | Manus | RS92 | 2 | 02.11.2012 | 00:00:00 | Yes | No | Night | 74 | Manus | RS92 | 2 | 23.11.2013 | 00:00:00 | Yes | Yes | Night |
| 37 | Manus | RS92 | 2 | 17.11.2012 | 00:00:00 | Yes | No | Night | 75 | Manus | RS92 | 2 | 28.11.2013 | 12:00:00 | Yes | Yes | Day |
| 38 | Manus | RS92 | 2 | 20.12.2012 | 00:00:00 | Yes | Yes | Night | 76 | Manus | RS92 | 2 | 02.12.2013 | 00:00:00 | Yes | No | Night |