# Peer review of "Consistency between GRUAN sondes, LBLRTM and IASI"

_Atmospheric Measurement Techniques, 2016_

## Referee Comment (RC1) · Anonymous Referee #1 · 13 Dec 2016

**GENERAL COMMENTS**

This is a concise paper that takes advantage of the high quality, and resolved measurement uncertainties, provided by the GCOS Reference Upper-Air Network (GRUAN) to demonstrate that top of the atmosphere radiances calculated using the LBLTRM radiative transfer model using GRUAN temperature and water vapour profiles as input, are consistent with top of the atmosphere outgoing infra-red radiances measured by IASI. The paper also provides some evidence that the water vapour values measured by GRUAN are dry by about 2.5%. The paper will be of interest to the readers of AMT and will be suitable for publication in AMT after the issues highlighted below have been addressed. The required changes are relatively minor.

**SPECIFIC COMMENTS**

[Figure]

Page 1, line 4: I thought that LBLRTM was a radiative transfer model so I don't see how it can constitute a reference measurement?

Page 1, line 7: This statement is a little misleading, and could even be confusing for readers, given that LBLRTM and IASI do not measure relative humidity in the upper troposphere.

Page 1, line 19-20: This sentence makes no sense. Do you mean that it is mandatory (and for whom?) to ensure that satellite radiance measurements are consistent with profile measurements of ECVs in the nadir air column below the satellite? If that's what you mean then perhaps that's what you should say.

Page 2, line 2: Performed independently of what?

Page 2, line 3: What, exactly, is assimilated into short and medium range weather forecasting models? It is not at all clear from what you have written.

Page 2, line 7: Do you mean the top of the atmosphere radiance measurements or do you mean retrieved temperature or water vapour profiles?

Page 2, line 13: What is the difference between reproducible and comparable in this context?

Page 2, line 22: Which measurement process? Do you mean the radiosonde measurement process?

Page 3, line 6: Should this be laboratory measurements of gas absorption *spectra*?

Page 3, line 9: I think that you will need to be more specific what you mean by 'keeping the chain of traceability unbroken'.

Page 3, line 12: A retrieval and data assimilation are two completely different things and it is incorrect to present them as somehow being equivalent. A retrieval typically uses optimal estimation to infer atmospheric state variables and/or trace gas concentrations from spectral measurements while data assimilation uses those measurements

to nudge a model to a state closer to the true atmosphere as it runs, typically using variational data assimilation or ensemble Kalman filter approaches.

Page 3, line 19: Do you rather mean an ill-constrained problem?

Page 3, line 21: I don't think that it is so much more convenient to perform the comparisons in radiance space but rather a most robust process performing the comparisons in radiance space.

Page 4, line 3: I assume that you describe somewhere how you have quantified the collocation uncertainty to obtain a value for sigma?

Page 4, line 12: Do you think that a sentence is required to describe the units for spectral measurements as brightness temperatures for readers who may be more used to measurements in W/mˆ-2 sr cmˆ-1?

Page 4, line 13: You say 'with a relatively high vertical resolution and high degree of accuracy' but just on the previous page you say 'very different atmospheric profiles can lead to the same radiances measured at the top of the atmosphere'. These two sentences appear to communicate very different messages.

Page 4, line 27: In what sense are the calculated spectra 'accurate'? Surely they're just what they are given the input temperature, water vapour and ozone (?) profiles?

Page 5, line 11: This is not the primary goal for GRUAN as stated in the GRUAN literature. As detailed in GCOS-112, the purpose of GRUAN is to: i) Provide long-term high quality climate records; ii) Constrain and calibrate data from more spatially-comprehensive global observing systems (including satellites and current radiosonde networks); and iii) Fully characterize the properties of the atmospheric column.

Page 6, line 10: Noting that you are using data at 1 km resolution for your cloud screening, it makes me wonder if and how you have accounted for the fact that the radiosonde drifts quite far from its launch location during its flight.

Table 2: There are many columns in Table 2 that are unnecessary since they all contain the same entry. Please delete them and include this constant information in the Table caption (if necessary).

Page 6, line 18: The usability for what purpose?

Page 6, line 22: The phrase 'when they are below 100 hPa' is ambiguous. Do you mean at pressures below 100 hPa or at altitudes below 100 hPa?

Page 6, line 24: Are the balloons thicker or larger? I always thought that the thickness was the same and the size changes, but I may be wrong.

Page 6, line 30: How are any discontinuities between the temperature and water vapour profiles obtained from GRUAN and those obtained from the ECMWF reanalyses dealt with?

Page 7, line 2: I am not sure what you mean by 'takes as practical the Hyland and Wexler (1983) curve'.

Page 7, line 5: Too noisy in what regard?

Page 7, line 6: If IASI measured radiances or retrievals are not sensitive to particular small scales in the vertical, then it should not be necessary to smooth the GRUAN profiles. How different would the results of this study be if the profiles are not smoothed? If it makes no difference, then it would be better not to smooth the profiles since it would be essentially unnecessary. If to does make a difference, it would be very interesting to know how and why?

Figure 2: I don't understand the large leftward excursion of the green trace in this figure. Is this the temperature trace or the dew point temperature trace? Did the ECMWF reanalyses assimilate the Manus Island radiosonde data?

Page 8, line 2: How many times is several times? I would have thought that for Monte Carlo it would have to be several hundred times?

Page 8, line 9: Do you mean 'perturbed by plus one GRUAN standard deviation'? This is not very clearly worded and may confuse some readers.

Page 8, line 27: Rather than just saying that the match is 'quite remarkable' can you provide a quantitative metric (maybe the k value) the describes how good the match is.

page 8, line 32: Can a value for sigma in equation 1 be provided?

Figure 6: Just to confirm, the dotted line in Figure 6 is the k=2 line correct? I think that this should be stated clearly somewhere.

Page 10, line 5: But isn't it possible that the GRUAN humidity measurements are affected by the sonde passing through clouds and that would not be picked up in your analysis?

GRAMMAR AND TYPOGRAPHICAL ERRORS

I understand that the author's first language is not English. There are a number of grammatical errors in the paper, only a few of which I have documented below, that will need to be fixed and I would encourage the authors to find someone to very carefully proof read this paper and correct these errors - they do tend to detract from the excellent quality of the science. I am very surprised that the co-authors, whose first language is English, consented to this paper being submitted in this state.

Page 1, line 9: Replace 'ECV' with 'ECVs'.

Page 1, line 14: Replace 'are shown' with 'is shown'.

Page 2, line 15: Replace 'are assured' with 'is assured'.

Page 2, line 28: Replace 'the accurate' with 'an accurate'.

Page 3, line 6: Replace 'perform its calculations' with 'perform their calculations'.

Page 4, line 19: The CNES acronym needs to be expanded.

Page 6, line 6: Replace 'leaving 76 clear cases' with 'leaving 76 clear sky cases' and

likewise a few lines later.

Page 7, line 13: Either 'These spectra' or 'This spectrum'. Likewise elsewhere. Spectrum is the singular and spectra is the plural.

Page 10, line 22: Replace 'will kill the consistency results' with 'will adversely affect the consistency results'.

———————————————————

---

## Referee Comment (RC2) · Anonymous Referee #2 · 25 Feb 2017

Uploaded as pdf

Please also note the supplement to this comment:
http://www.atmos-meas-tech-discuss.net/amt-2016-344/amt-2016-344-RC2-supplement.pdf

---

## Author Comment (AC1)

**First of all I would like to thank the referees for their helpful comments, which I highly appreciate and will certainly make the paper more readable. Answers to the referee's comments are written in boldface font below.**

Anonymous Referee #1

GENERAL COMMENTS
This is a concise paper that takes advantage of the high quality, and resolved measurement uncertainties, provided by the GCOS Reference Upper-Air Network (GRUAN) to demonstrate that top of the atmosphere radiances calculated using the LBLTRM radiative transfer model using GRUAN temperature and water vapour profiles as input, are consistent with top of the atmosphere outgoing infra-red radiances measured by IASI. The paper also provides some evidence that the water vapour values measured by GRUAN are dry by about 2.5%. The paper will be of interest to the readers of AMT and will be suitable for publication in AMT after the issues highlighted below have been addressed. The required changes are relatively minor.
SPECIFIC COMMENTS

Page 1, line 4: I thought that LBLRTM was a radiative transfer model so I don't see how it can constitute a reference measurement?
**This is not clear in the paper, LBLRTM is a reference RTM. This will be clarified in the text. The new sentence will read: "This result is key to have consistency between radiosonde and satellite measurements for climate data records, since GRUAN, IASI and LBLRTM constitute reference measurements or radiative transfer models in each of their fields".**

Page 1, line 7: This statement is a little misleading, and could even be confusing for readers, given that LBLRTM and IASI do not measure relative humidity in the upper Troposphere.
**IASI does measure relative humidity in the upper troposphere and, although not with a very high vertical resolution, it is very sensitive to it.**

Page 1, line 19-20: This sentence makes no sense. Do you mean that it is mandatory (and for whom?) to ensure that satellite radiance measurements are consistent with profile measurements of ECVs in the nadir air column below the satellite? If that's what you mean then perhaps that's what you should say.
**Agreed. Sentence will be changed to: "To measure the ECVs it is necessary to convert measured radiances into atmospheric temperature and water vapor profiles"**

Page 2, line 2: Performed independently of what?
**Agreed. Sentence will be changed to:**

**"This inversion can either be performed as a straight forward inversion, or in the case of Numerical Weather Prediction (NWP), by assimilating radiances into short or medium range forecasting models"**

Page 2, line 3: What, exactly, is assimilated into short and medium range weather forecasting models? It is not at all clear from what you have written.

**See answer above.**
Page 2, line 7: Do you mean the top of the atmosphere radiance measurements or do you mean retrieved temperature or water vapour profiles?
**Sentence will be changed to:**
**"Whether radiances or temperature and water vapor profiles are measured, for them to be useful for climate or any other application, they need to be adequately calibrated"**
Page 2, line 13: What is the difference between reproducible and comparable in this Context?
**Reproducible means that the experiment or measurement can be repeated many times and still gives a similar result. Comparable means that measurements can be compared.**
Page 2, line 22: Which measurement process? Do you mean the radiosonde measurement Process?
**The sentence seems a bit isolated from the rest. The last sentence from the previous paragraph will be separated from that paragraph and it will be attached to this paragraph. The paragraph will then start with:**
**"For the case described here, the measurement process**
**consists of three fundamental elements; i) the radiance measurement from the satellite instrument, ii) the temperature and water**
**vapour measurements from the radiosondes and iii) the RTM that establishes the link between them.**
**Throughout this measurement process, not all elements in the traceability chain are usually used comprehensively."**
Page 3, line 6: Should this be laboratory measurements of gas absorption *spectra*?
**Yes, this will be added in the text: "They make use of laboratory measurements of gas absorption spectra to perform its calculations,"**
Page 3, line 9: I think that you will need to be more specific what you mean by 'keeping the chain of traceability unbroken'.
**The reference from Disksen et al. 2014 will be introduced here.**
Page 3, line 12: A retrieval and data assimilation are two completely different things and it is incorrect to present them as somehow being equivalent. A retrieval typically uses optimal estimation to infer atmospheric state variables and/or trace gas concentrations from spectral measurements while data assimilation uses those measurements to nudge a model to a state closer to the true atmosphere as it runs, typically using variational data assimilation or ensemble Kalman filter approaches.
**They are similar in the sense that they both transform radiances into atmospheric profiles of temperature and water vapor. Sentence will be changed to: "When**

**transforming IR measured radiances into atmospheric parameters, effectively performing what are known as retrievals, or as a component of data assimilation where radiances are used to improve the original atmospheric profile estimation,"**

Page 3, line 19: Do you rather mean an ill-constrained problem?

**It is an ill-posed problem, according to the mathematical definition: https://www.encyclopediaofmath.org/index.php/Ill-posed_problems**

Page 3, line 21: I don't think that it is so much more convenient to perform the comparisons in radiance space but rather a most robust process performing the comparisons in radiance space.

**Agreed. Sentence changed to: "It is therefore a much more robust process to perform the comparisons in radiance space, where the problem is uniquely determined (e.g. Calbet et al. (2011))."**

Page 4, line 3: I assume that you describe somewhere how you have quantified the collocation uncertainty to obtain a value for sigma?

**This is still an open problem. This paper does not deal with it, but rather the comparison of IASI and GRUAN.**

Page 4, line 12: Do you think that a sentence is required to describe the units for spectral measurements as brightness temperatures for readers who may be more used to measurements in W/m^-2 sr cm^-1?

**Historically, units of K and W/m-2 sr cm-1 are used interchangeably in the noise figures within the satellite community, so there should be no problem in mentioning different units.**

Page 4, line 13: You say 'with a relatively high vertical resolution and high degree of accuracy' but just on the previous page you say 'very different atmospheric profiles can lead to the same radiances measured at the top of the atmosphere'. These two sentences appear to communicate very different messages.

**That is why the word "relatively" is used. Although it is an ill-posed problem, the IASI retrievals do have a vertical resolution and accuracy which are finite.**

Page 4, line 27: In what sense are the calculated spectra 'accurate'? Surely they're just what they are given the input temperature, water vapour and ozone (?) profiles?

**Agreed. Sentence changed to: "Spectra at the top of the atmosphere were generated using the reference Line By Line Radiative Transfer Model (LBLRTM, Clough et al. (2005))"**

Page 5, line 11: This is not the primary goal for GRUAN as stated in the GRUAN literature. As detailed in GCOS-112, the purpose of GRUAN is to: i) Provide longterm high quality climate records; ii) Constrain and calibrate data from more spatiallycomprehensive global observing systems (including satellites and current radiosonde networks); and iii) Fully characterize the properties of the atmospheric column.

**Agreed. Sentence changed to "Two of GRUAN's primary goal is to fully characterize the properties of the atmospheric column and their changes"**

Page 6, line 10: Noting that you are using data at 1 km resolution for your cloud screening, it makes me wonder if and how you have accounted for the fact that the radiosonde drifts quite far from its launch location during its flight.

**IASI and GRUAN collocation is not the main subject of this paper. Same is true for cloud detection. Although the best effort has been taken to avoid collocation problems or cloud detection issues, it is clear that there is always the chance that there might be some remaining undesired effects here (wrong collocation or undetected clouds). To overcome the cloud detection issue, a visual inspection of the scene has been made to be more certain the scene is clear.**

Table 2: There are many columns in Table 2 that are unnecessary since they all contain the same entry. Please delete them and include this constant information in the Table caption (if necessary).

**Agreed. Table will be modified accordingly.**

Page 6, line 18: The usability for what purpose?

**I would say for any purpose.**

Page 6, line 22: The phrase 'when they are below 100 hPa' is ambiguous. Do you mean at pressures below 100 hPa or at altitudes below 100 hPa?

**I do not believe so. Below means physically below. This is emphasized by the next sentence which says below the tropopause. Wording will be changed to "physically below" to avoid any ambiguities.**

Page 6, line 24: Are the balloons thicker or larger? I always thought that the thickness was the same and the size changes, but I may be wrong.

**Thickness and size at burst are related. A thicker balloon bursts higher than a thinner one because they can reach a bigger size without exploding.**

Page 6, line 30: How are any discontinuities between the temperature and water vapour profiles obtained from GRUAN and those obtained from the ECMWF reanalyses dealt With?

**Sentence added to clarify this: "In the sample dealt with in this paper there are no big discontinuities between GRUAN measurements and ECMWF profiles, therefore no measure has been taken to smoothen the profiles at the intersection point."**

Page 7, line 2: I am not sure what you mean by 'takes as practical the Hyland and Wexler (1983) curve'.

**Agreed. Changes "as practical" to "the one best representing the reality".**

Page 7, line 5: Too noisy in what regard?

**Changing "too noisy" for "extremely oscillatory".**

Page 7, line 6: If IASI measured radiances or retrievals are not sensitive to particular small scales in the vertical, then it should not be necessary to smooth the GRUAN profiles. How different would the results of this study be if the profiles are not smoothed? If it makes no difference, then it would be better not to smooth the profiles since it would be essentially unnecessary. If to does make a difference, it would be very interesting to know how and why?

**The smoothing is done on a very small vertical scale compared to IASI vertical resolution, so it will have very little impact on the results. Nevertheless, the smoothing is necessary as to take some average value of the GRUAN measurement in the vertical and not some spike or random oscillation.**

Figure 2: I don't understand the large leftward excursion of the green trace in this

figure. Is this the temperature trace or the dew point temperature trace? Did the ECMWF reanalyses assimilate the Manus Island radiosonde data?

**This big excursion is coming from the humidity measurement. It is a good example of a spike in the data, which is almost certainly unreal and might be coming from, for example, a cosmic ray impinging on the humidity sensor or the electronics.**

**As far as I know, these are "regular" or synoptic radiosonde data, se they have been most likely assimilated into ECMWF.**

Page 8, line 2: How many times is several times? I would have thought that for Monte Carlo it would have to be several hundred times?

**Agreed that better wording is needed. Changed the sentence to:**

**"This process is repeated several times to obtain the standard deviation of the radiances within the Monte Carlo approach. Since only an estimation of the standard deviation is needed, not too many repetitions are necessary, and with 15 repetitions a standard deviation with a sufficiently small uncertainty was obtained."**

Page 8, line 9: Do you mean 'perturbed by plus one GRUAN standard deviation'? This is not very clearly worded and may confuse some readers.

**I believe it is clearer to say plus one GURAN uncertainty, because GRUAN provides uncertainties, not standard deviations**

Page 8, line 27: Rather than just saying that the match is 'quite remarkable' can you provide a quantitative metric (maybe the k value) the describes how good the match is.

**The sentence will be changed to "visually, the match is quite remarkable".**

**A k value would be nice to give, but we cannot estimate it precisely because we do not know all the elements from Eq. 1. We know the IASI instrument uncertainty quite well, but we only know the GRUAN uncertainty in radiance space partially (because we are lacking the uncertainty covariance matrix) and we certainly do not know the collocation uncertainty (sigma). This will be explained better in the conclusions of the paper.**

page 8, line 32: Can a value for sigma in equation 1 be provided?

**No, the main purpose of this paper is not to estimate the sigma (collocation) uncertainty**

Figure 6: Just to confirm, the dotted line in Figure 6 is the k=2 line correct? I think that this should be stated clearly somewhere.

**Agreed. This is confusion. Figure caption and main text will be changed.**

Page 10, line 5: But isn't it possible that the GRUAN humidity measurements are affected by the sonde passing through clouds and that would not be picked up in your Analysis?

**It is always possible that some undetected cloud is present in some samples. But this is highly unlikely, for several reasons. One of them is the visual inspection of clouds that has been done on all the samples. Another one is that the match is quite good, if there were some cloud contamination this would not have been possible.**

GRAMMAR AND TYPOGRAPHICAL ERRORS

I understand that the author's first language is not English. There are a number of grammatical errors in the paper, only a few of which I have documented below, that will need to be fixed and I would encourage the authors to find someone to very carefully proof read this paper and correct these errors - they do tend to detract from the

excellent quality of the science. I am very surprised that the co-authors, whose first language is English, consented to this paper being submitted in this state.
Page 1, line 9: Replace 'ECV' with 'ECVs'.
Page 1, line 14: Replace 'are shown' with 'is shown'.
Page 2, line 15: Replace 'are assured' with 'is assured'.
Page 2, line 28: Replace 'the accurate' with 'an accurate'.
Page 3, line 6: Replace 'perform its calculations' with 'perform their calculations'.
Page 4, line 19: The CNES acronym needs to be expanded.
Page 6, line 6: Replace 'leaving 76 clear cases' with 'leaving 76 clear sky cases' and likewise a few lines later.
Page 7, line 13: Either 'These spectra' or 'This spectrum'. Likewise elsewhere. Spectrum is the singular and spectra is the plural.
Page 10, line 22: Replace 'will kill the consistency results' with 'will adversely affect the consistency results'.

Anonymous Referee #2
Review of "Consistency between GRUAN sondes, LBLRTM and IASI" by Xavier Calbet and colleagues

Calbet et al. perform an attempt at metrological closure of a satellite-to-sonde comparison. The methods and conclusions drawn are sound. As such this should be published in AMT. I have a number of queries and text suggestions which the authors should consider in revisions which I detail below.

Questions

1. When noting the distinction between the Lindenberg characterization and your results (p.5, lines 22-27) presumably this may be due to geographical gradients in spatio-temporal scales of the ECVs. Specifically, in the TWP there is frequent convection which may lead to far lower spatio-temporal coherence of fields than in the mid-latitudes where mixing tends to be dominated by synoptic and not mesoscale features. It would help the reader to discuss this issue a little more thoroughly in such a context if possible.

**Thanks for the comment. This note will be included in the text. The sentence will be changed to: "Pougatchev et al. (2009) studied the variability of temperature and water vapour with radiosondes launched from Lindenberg reaching the conclusion that to minimize the collocation uncertainty a spatial and temporal window 25 of 25 km and 30 minutes respectively is needed. This collocation criteria is clearly derived from a particular station on the globe and it might not extrapolate well to other regions, particularly to the Tropical Western Pacific region in which Manus is. Nevertheless this figure can be taken as a first approximation. During the development of this study it was noted that for water vapour these criteria in reality seem to be too relaxed. Therefore,**

**even stricter criteria are needed (see section 4 and 5 for a discussion on this)."**

2. I'd like to see an attempt to quantify the error use of the ECMWF fields above 100hPa could possibly have (p. 6 lines 14-30). Presumably the potential effect is calculable and it would aid the reader if this could be clearly articulated here.
**I would find difficult to estimate this error. The only thing we can say is that the match is moderately good, Fig 6 and 8, and that it seems better for ECMWF data than for GRUAN (+ ECMWF on the higher levels)**

3. I would interpret the GRUAN profile smoothing on p.7 lines 4 to 8 as being an attempt to minimize the mismatch error term sigma in equation 1. The authors should explicitly state this here I think.
**No. The smoothing is done for the purpose of avoiding the (natural) oscillations in the measurements and to remove the spikes present in the data. See comments to Referee #1 to see how the final text would be.**

4. On page 9 paragraph starting on line 4 it is unclear whether the authors are closing this at 2 sigma (k=2) as implied by the text or 1 sigma (k=1 as given). Please clarify and avoid ambiguity here.
**A similar comments has been raised by another colleague. This whole paragraph will be reworded to be more precise.**

5. I'd like the authors to consider giving some potential avenues to investigate on 1800cm-1 to 1840cm-1 on page 9. They must have some vague notions at least as to what could plausibly explain this and hence where further investigation on the matter should be directed.
**No avenue yet. This would be the subject of research of another paper. Once we estimate the GRUAN uncertainties in radiance space well (partly done in this paper, but to fully do this, the full uncertainty covariance matrix for GRUAN is needed) or the collocation uncertainty (no attempt to estimate this in this paper, again the subject of another paper) we should be able to investigate more in depth why some wavenumber deviate.**

6. The paragraph starting line 29 of page 9 feels like it would make more sense placed elsewhere to me. Perhaps where the subsampling is already discussed would be more appropriate than here?
**You are probably right, but I am not sure where to move it to.**

7. I'd like to see a recognition that the GRUAN product may be modified to account for the residual daytime effect in the text (assuming this is the intention).
**This is highlighted in the abstract and the conclusions. This is probably enough. Unless you mean something else.**

Suggestions

1. Abstract line 3 I would remove "to have consistency between radiosonde and satellite measurements" as this is already implicit from the prior text.
 2. Page 1, line 17 suggest [...] several advantages, in particular: i) [...]

3. Page 1, line 18 suggest [...] but rather indirectly sense it by [...]

4. Page 2, line 3 [...](NWP), radiances are [...]

5. Page 2 line 4 [...] inaccuracies due to one or more of: i) [...]

6. Page 2 line 6 delete or others or specify what these are

7. Page 2 line 7 [...]climate or many other applications [...]

8. Page 4 line 4 [...] of the equation. For [...]

9. Page 5 line 14 [...] certified as meeting GRUAN [...]

10. Page 6, line 11 [...]the remainder of this paper. [...]

11. Page 7, line 24 [...]multiplied by the[...]

12. Page 7, line 32 remove the parentheses

13. Page 8, line 17 [...]as would be expected.

14. Page 8, line 34 remove 'resides'

15. Page 9, line 20 [...]lying in some parts[...]

16. Page 10, line 2 [...]each other, especially for night-time ascents. [...]